# Brd4 binds to active enhancers to control cell identity gene induction in adipogenesis and myogenesis

Ji-Eun Lee [1], Young-Kwon Park[1], Sarah Park[1], Younghoon Jang[1], Nicholas Waring[1,2], Anup Dey[3], Keiko Ozato[3], Binbin Lai[1,2], Weiqun Peng[2] & Kai Ge [1]

The epigenomic reader Brd4 is an important drug target for cancers. However, its role in cell differentiation and animal development remains largely unclear. Using two conditional knockout mouse strains and derived cells, we demonstrate that Brd4 controls cell identity gene induction and is essential for adipogenesis and myogenesis. Brd4 co-localizes with lineage-determining transcription factors (LDTFs) on active enhancers during differentiation. LDTFs coordinate with H3K4 mono-methyltransferases MLL3/MLL4 (KMT2C/KMT2D) and H3K27 acetyltransferases CBP/p300 to recruit Brd4 to enhancers activated during differentiation. Brd4 deletion prevents the enrichment of Mediator and RNA polymerase II transcription machinery, but not that of LDTFs, MLL3/MLL4-mediated H3K4me1, and CBP/p300-mediated H3K27ac, on enhancers. Consequently, Brd4 deletion prevents enhancer RNA production, cell identity gene induction and cell differentiation. Interestingly, Brd4 is dispensable for maintaining cell identity genes in differentiated cells. These findings identify Brd4 as an enhancer epigenomic reader that links active enhancers with cell identity gene induction in differentiation.

[1] Adipocyte Biology and Gene Regulation Section, Laboratory of Endocrinology and Receptor Biology, National Institute of Diabetes and Digestive and Kidney Diseases, National Institutes of Health, Bethesda, MD 20892, USA. [2] Departments of Physics and Anatomy and Regenerative Biology, The George Washington University, Washington, DC 20052, USA. [3] Program in Genomics of Differentiation, National Institute of Child Health and Human Development, National Institutes of Health, Bethesda, MD 20892, USA. Correspondence and requests for materials should be addressed to K.G. (email: kai.ge@nih.gov)

D ifferentiation is controlled by cell-type-specific gene expression, which is under the control of transcriptional enhancers[1]. Enhancers possess recognition motifs for sequence-specific lineage-determining transcription factors (LDTFs), which bind to and activate enhancers[2]. LDTFs recruit epigenomic regulators that remodel the chromatin landscape by adding epigenetic modifications (i.e., methylation, acetylation, etc.) to the histone tails of the associated nucleosomes, after which RNA polymerase II (Pol II) is recruited and transcription of enhancer RNA (eRNA) and nearby genes occurs[2]. Enhancers are enriched with histone H3K4 mono-methylation (H3K4me1), which is mainly deposited by H3K4 methyltransferases MLL3 (KMT2C) and MLL4 (KMT2D)[3]. H3K4me1 precedes the addition of the active enhancer mark H3K27ac by histone acetyl-transferases CBP/p300[4,5]. CBP/p300 binding identifies active enhancers that control cell-type-specific gene expression[6]. The acetyl marks on histones act as docking sites for epigenomic readers that have conserved bromodomains[7].

The epigenomic reader Brd4 is a member of the bromodomain and extra-terminal domain (BET) family of nuclear proteins that also include Brd2 and Brd3[8,9]. Brd4 is enriched on active enhancers and promoters[10,11]. Cooperation between LDTFs and CBP/p300 facilitates Brd4 recruitment to its target promoters and enhancers[11]. Small-molecule competitive BET inhibitors such as JQ1 bind to the acetyl-lysine binding pocket of the BET bro-modomains and displaces BET proteins from chromatin[12]. JQ1-mediated inhibition of Brd4 binding leads to the displacement of the Mediator complex and Pol II from enhancers, which in turn reduces eRNA production and associated gene expression[10,13,14].

Adipogenesis and myogenesis are two model systems that are well suited for studying cell differentiation. In adipogenesis, the induction of early adipogenic transcription factors (TFs), including CCAAT/enhancer binding protein-β (C/EBPβ), in turn induces the expression of two master adipogenic TFs, peroxisome proliferator-activated receptor-γ (PPARγ) and C/EBPα[15,16]. PPARγ and C/EBPα work in cooperation to activate many adipocyte-specific genes. The synchronized nature of adipogenesis in cell culture, wherein the majority of the cells in the confluent population differentiate from preadipocytes to mature adipocytes within 6–8 days, allows for a system conducive to studying gene expression during the process of differentiation[17]. Myogenesis is another model of synchronized cell differentiation. Myogenic differentiation protein (MyoD) and myogenic factor 5 (Myf5) are required for commitment to the muscle differentiation program, while myogenin plays a necessary role in establishing the terminal muscle phenotype[18].

Brd4 is a prominent drug target for cancers but its role in normal cell differentiation and tissue development is largely unexplored. In this study, we use adipogenesis and myogenesis as model systems to explore the role of Brd4 in differentiation and development. Using tissue-specific Brd4 KO mice, we provide in vivo evidence that Brd4 is essential for adipogenesis and myogenesis. During adipogenesis, Brd4 preferentially binds to active enhancers and controls the induction of cell-type-specific genes. Furthermore, we determine the sequential actions of LDTFs, enhancer epigenomic writers MLL3/MLL4 and CBP/p300, Brd4, transcription coactivator complex Mediator and general transcription factor TFIID, Pol II, and transcription

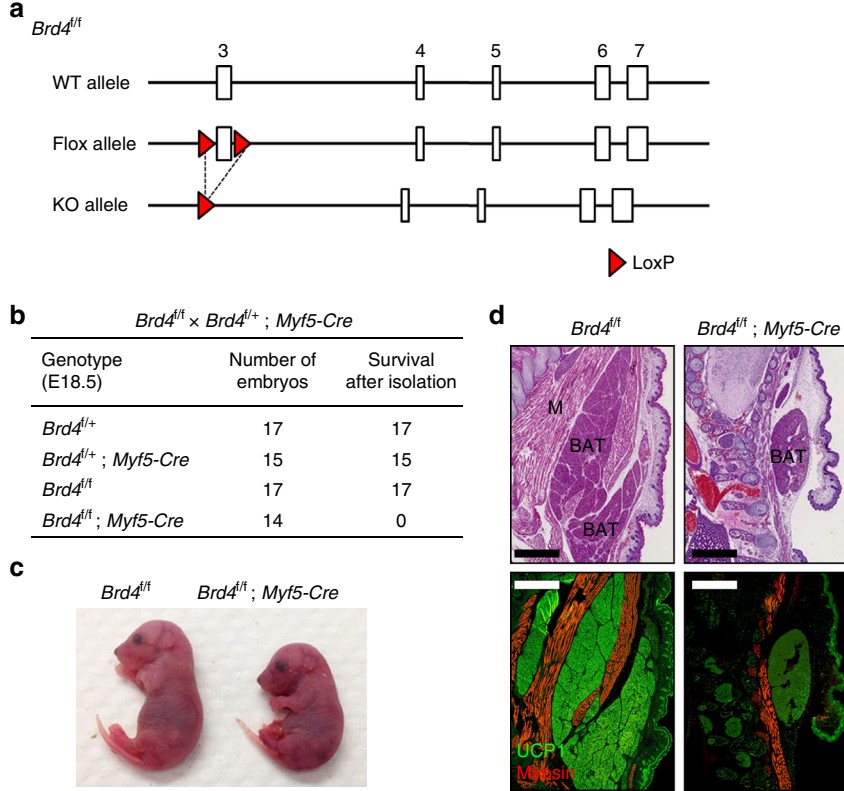

**Fig. 1** Brd4 is required for the development of brown adipose tissue and muscle. Brd4 f/f mice were crossed with Myf5-Cre mice. **a** Schematics of wild-type (WT) allele, conditional KO (flox) allele, and KO allele of Brd4 f/f mice. In the flox allele, exon 3 was flanked by two loxP sites. **b** Genotype of progeny from crossing between Brd4 f/f;Myf5-Cre and Brd4 f/f mice at E18.5. Brd4 f/f;Myf5-Cre mice survived at E18.5 but died within minutes after cesarean section due to muscle defects around rib cage. Similar phenotypes were observed in 14 Brd4 f/f;Myf5-Cre mice. **c** Representative morphology of E18.5 embryos. **d** Histological analysis of E18.5 embryos. Sagittal sections of cervical/thoracic area were stained with H&E (upper panels) or with antibodies against the brown adipose tissue (BAT) marker UCP1 (green) and the muscle (M) marker myosin (red) (lower panels). Scale bars = 80 μm

elongation factor p-TEFb in enhancer activation and cell identity gene induction during differentiation. We further show that unlike its critical role in cell differentiation, Brd4 is largely dispensable for the maintenance of cell identity gene expression in differentiated cells.

## Results

**Brd4 is required for adipose tissue and muscle development.** Two independently developed *Brd4* conditional knockout (KO) mouse strains, *Brd4*[f/f] and *Brd4*[f/f #2], were used in this study. In the *Brd4*[f/f] strain, the exon 3 was flanked by two loxP sites (Fig. 1a), while in the *Brd4*[f/f #2] strain, the exon 5 was floxed (Supplementary Fig 1a). To study the role of Brd4 in adipose tissue and muscle development, we crossed *Brd4*[f/f] with *Myf5-Cre* mice to delete *Brd4* gene in progenitor cells of brown adipose tissue (BAT) and muscle lineages[19,20]. The resulting *Brd4*[f/f];*Myf5-*

*Cre* mice survived until birth. E18.5 *Brd4*[f/f];*Myf5-Cre* embryos were obtained at the expected Mendelian ratio but were unable to breathe and died immediately (Fig. 1b). They showed an abnormal hunched posture due to severe reduction of back muscles (Fig. 1c). Immunohistochemical analysis of cervical regions of E18.5 embryos revealed that the deletion of *Brd4* leads to severe reduction of BAT and muscle mass, indicating that Brd4 is essential for BAT and muscle development (Fig. 1d).

**Brd4 controls cell identity gene induction.** To investigate how Brd4 regulates adipose tissue development, we isolated primary *Brd4*[f/f] preadipocytes from BAT. After immortalization, cells were infected with adenoviruses expressing green fluorescent protein (GFP) or Cre (Fig. 2a, b and Supplementary Fig 2). Deletion of *Brd4* by Cre did not affect the growth rate of SV40T-immortalized brown preadipocytes (Fig. 2c), but prevented adipogenesis and the induction of adipocyte marker genes such as *Pparg*, *Cebpa*, and *Fabp4* (Fig. 2d, e). We confirmed the essential role of Brd4 in adipogenesis in an independent brown preadipocyte cell line derived from *Brd4*[f/f #2] mice (Supplementary Fig 1b–d). Consistent with the phenotypes observed in both *Brd4* knockout cell lines, knockdown of *Brd4* in 3T3-L1 white preadipocytes impaired adipogenesis (Supplementary Fig 3). Knockdown of *Brd4* in C2C12 myoblast inhibited myogenesis and myocyte gene expression (Supplementary Fig 4a–d).

To find out how Brd4 regulates gene expression during adipogenesis, we performed RNA-Seq analyses before (day 0, D0) and during (D2) adipogenesis of *Brd4*[f/f] preadipocytes infected with adenoviral GFP or Cre. Using a 2.5-fold cutoff for differential expression, we defined up-regulated (816/16,323), down-regulated (1045/16,323), and unchanged (14,462/16,323) genes from D0 to D2 of differentiation (Fig. 2f). Among the 816 up-regulated genes, 351 were induced in a Brd4-dependent manner. Interestingly, only Brd4-dependent up-regulated genes were strongly associated functionally with fat cell differentiation and lipid metabolism (Fig. 2g). During C2C12 myogenesis, Brd4-dependent up-regulated genes were preferentially associated functionally with muscle development and function (Supplementary Fig 4e, f). These results suggest that Brd4 controls cell identity gene induction during adipogenesis and myogenesis.

**Brd4 binds to cell identity genes.** Next, we performed ChIP-Seq to map the genomic binding of Brd4 before (D0), during (D2), and after (D7) adipogenesis. To exclude false-positive genomic

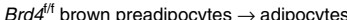

Brd4[f/f] brown preadipocytes → adipocytes

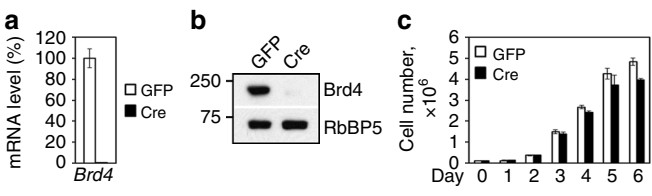

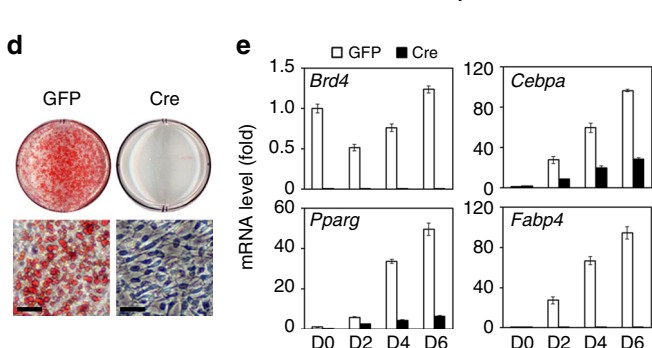

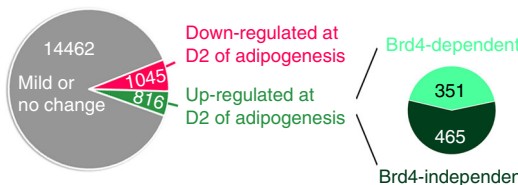

**Fig. 2** Brd4 controls cell identity gene induction during adipogenesis. **a–e** Brd4 is required for adipogenesis of brown preadipocytes. SV40T-immortalized *Brd4*[f/f] brown preadipocytes were infected with adenoviral GFP or Cre, followed by adipogenesis assay. **a** qRT-PCR confirmation of *Brd4* deletion in preadipocytes. **b** Deletion of Brd4 was confirmed by western blot analysis. RbBP5 was used as a loading control. **c** Deletion of *Brd4* has little effect on the growth of immortalized preadipocytes. **d** Oil red O staining at D6 of adipogenesis. Upper panels, stained dishes; lower panels, representative fields under microscope. Scale bars = 15 μm. **e** qRT-PCR of *Brd4*, *Pparg*, *Cebpa*, and *Fabp4* expression at indicated time points during adipogenesis. D0/2/4/6, day 0/2/4/6. The data are presented as means ± SD. Three technical replicates from a single experiment were used. **f**, **g** Brd4 is required for induction of cell identity genes during adipogenesis. Cells were collected before (D0) and during (D2) adipogenesis for RNA-Seq. (f) Schematic of identification of Brd4-dependent and -independent up-regulated genes during adipogenesis. The cutoff for up- or down-regulation is 2.5-fold. The cutoff for Brd4 dependency is 2.5-fold. **g** Gene ontology (GO) analysis of gene groups defined in **f**

| Gene group | GO term | P value |
|---|---|---|
| Up-regulated (Brd4-dependent) | Brown fat cell differentiation | 7.8E−11 |
| | Fat cell differentiation | 3.6E−10 |
| | Triglyceride metabolic process | 6.7E−06 |
| | Oxidation reduction | 8.1E−06 |
| | Lipid metabolic process | 1.3E−05 |
| | Acylglycerol metabolic process | 2.0E−05 |
| | Glycerol ether metabolic process | 2.6E−05 |
| | Neutral lipid metabolic process | 2.6E−05 |
| | Fatty acid metabolic process | 3.2E−05 |
| Up-regulated (Brd4-independent) | Regulation of development | 1.8E−05 |
| | Cellular response to hormone | 2.2E−05 |
| | Intracellular signaling cascade | 3.2E−05 |
| Down-regulated | Cell adhesion | 2.3E−07 |
| | Regulation of transferase activity | 8.4E−05 |
| | Regulation of kinase activity | 1.2E−04 |
| | Intracellular signaling cascade | 2.4E−04 |

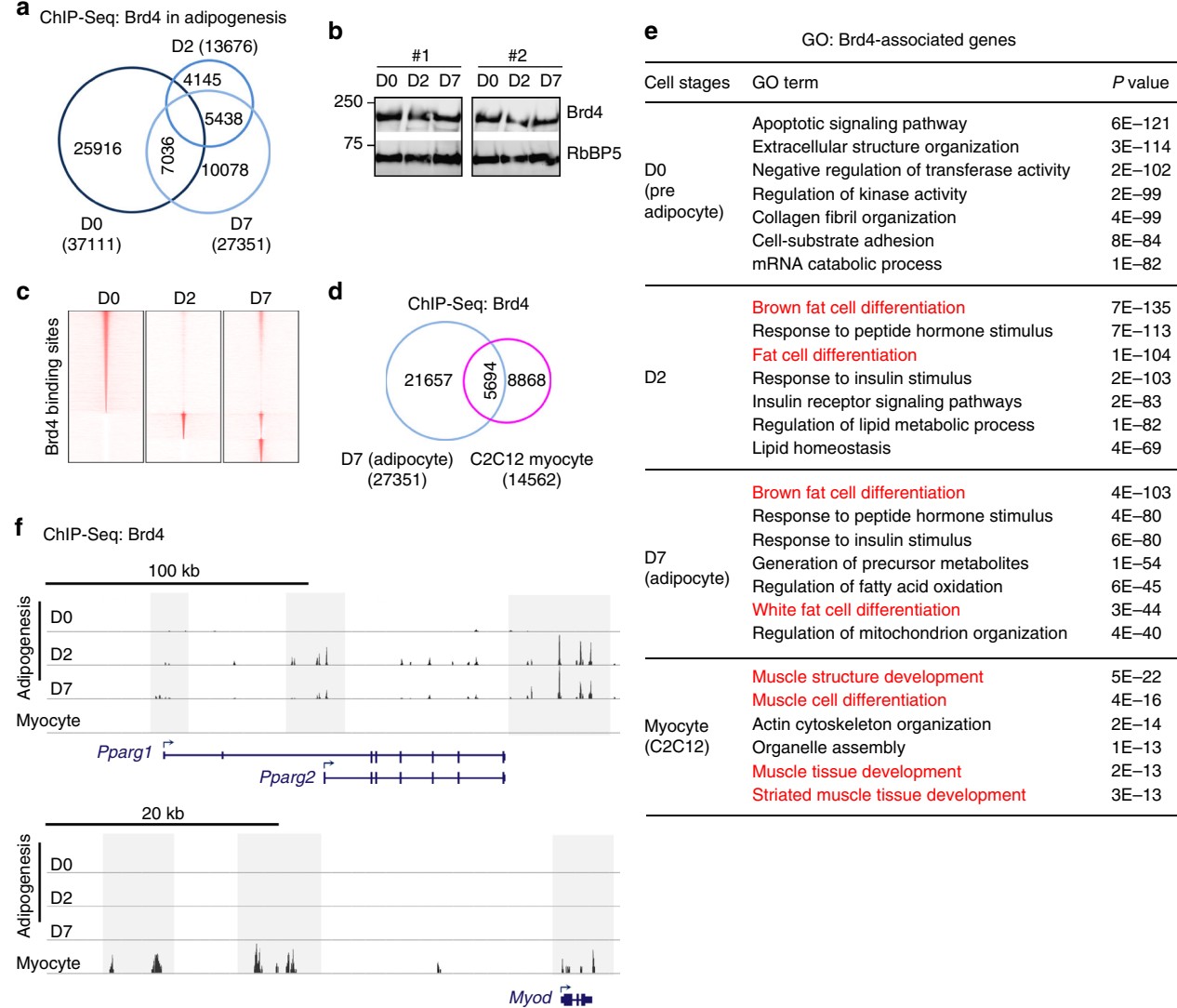

**Fig. 3** Cell-type- and differentiation-stage-specific binding of Brd4 on cell identity genes. Adipogenesis and myogenesis were done as in Fig. 2 and Supplementary Fig. 4, respectively. Cells were collected for ChIP-Seq analysis of Brd4. **a** Venn diagram depicting high-confidence Brd4 binding regions at D0 (preadipocytes), D2 (during adipogenesis), and D7 (adipocytes) of adipogenesis. High-confidence Brd4 binding regions were obtained by selecting overlapping regions from two biological replicates. **b** Western blot analysis of Brd4 during adipogenesis using nuclear extracts prepared from two independent brown preadipocyte cell lines. **c** Heat maps of Brd4 binding sites during adipogenesis. **d** Venn diagram representing Brd4 binding regions in adipocytes and myocytes. **e** GO analysis of genes associated with Brd4 binding regions at indicated time points and cell types. **f** ChIP-Seq profiles of Brd4 binding on gene loci encoding PPARγ and MyoD at indicated time points and cell types

Brd4 binding sites due to off-target effect of the antibody, ChIP-Seq was also done in *Brd4* KO cells. To identify high-confidence Brd4 binding sites, we first removed false-positive signals obtained from *Brd4* KO cells and then selected overlapping peaks from two biological replicates. In total, we identified 37,111, 13,676, and 27,351 Brd4 binding regions at D0, D2, and D7, respectively (Fig. 3a). Brd4 protein levels decreased only mildly at D2 (Fig. 3b), but the genomic binding of Brd4 redistributed dramatically from D0 to D2 (Fig. 3c). Around 89% of D0 Brd4 binding regions were lost at D2, although some of the lost regions were re-occupied by Brd4 at D7. We also performed Brd4 ChIP-Seq in C2C12 myocytes. The Brd4 binding regions in adipocytes and C2C12 myocytes were largely non-overlapping (Fig. 3d).

To characterize Brd4-associated genes in different stages and cell types, we assigned each Brd4 binding region to the nearest annotated gene. Gene ontology (GO) analysis revealed that at D0 before differentiation, Brd4 binds to genes associated with general

biological functions. However, Brd4 moves to adipogenesis-related genes at D2 and D7 of differentiation (Fig. 3e). In C2C12 myocytes, preferred target genes of Brd4 were those involved in muscle cell differentiation. Accordingly, we observed differentiation-stage- and cell-type-specific genomic binding of Brd4 on *Pparg* and *Myod1* loci, which encode the master adipogenic TF PPARγ and myogenic TF MyoD, respectively (Fig. 3f). Together, these results indicate cell-type- and differentiation-stage-specific genomic binding of Brd4 on cell identity genes.

**Brd4 co-localizes with LDTFs on active enhancers.** Next, we performed motif analysis of the top 3000 Brd4 binding regions at each time point and in different cell types (Fig. 4a). In pre-adipocytes (D0), Brd4 binding regions were enriched with motifs of AP-1 family of TFs Jun, Jdp2, and JunD. This is consistent with the previous finding that Brd4 interacts directly with c-Jun[21]. During (D2) and after (D7) adipogenesis, Brd4 binding regions

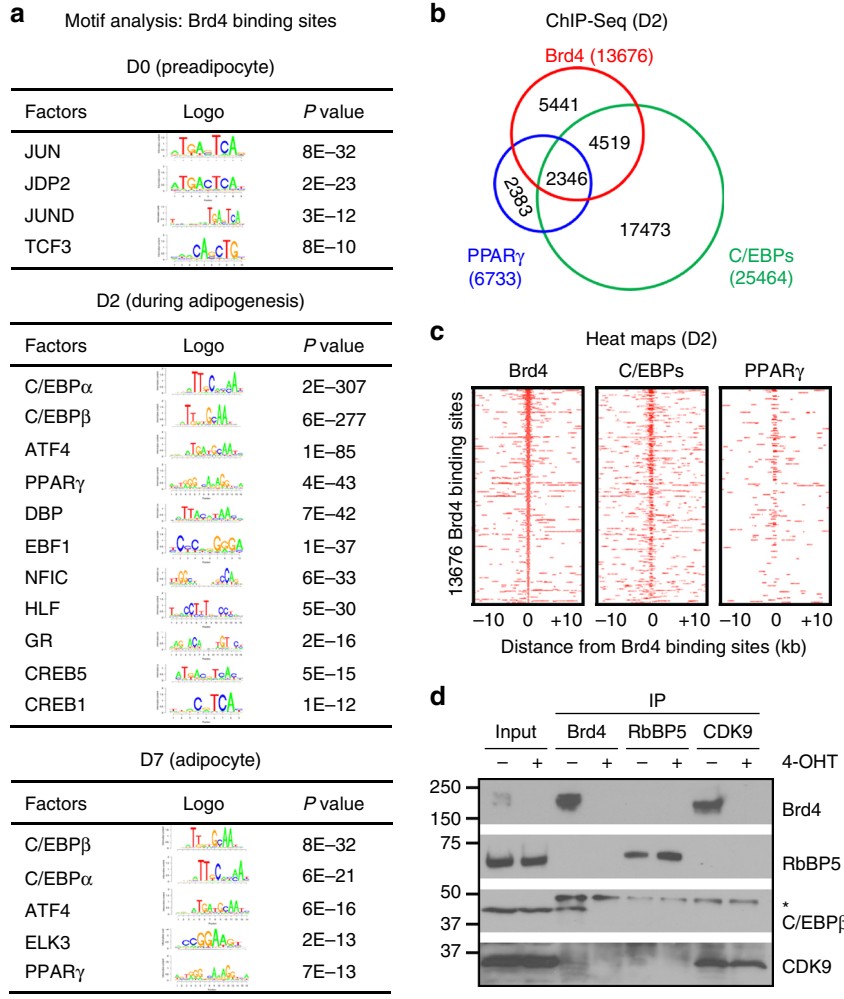

**Fig. 4** Genomic co-localization of Brd4 with LDTFs during adipogenesis. **a** Motif analysis of Brd4 binding regions in adipogenesis. Only TFs that are expressed at the indicated cell stages are presented. **b**, **c** Venn diagram **b** and heat maps **c** of genomic co-localization of Brd4 with C/EBPs (C/EBPα or β) and PPARγ at D2 of adipogenesis. Published ChIP-Seq data sets for C/EBPα/β and PPARγ were used (GSE74189)[24]. **d** Brd4 physically interacts with C/EBPβ during adipogenesis. SV40T-immortalized Brd4[f/f #2];Cre-ER brown preadipocytes were treated with 4-hydroxytamoxifen (4-OHT) to induce the deletion of exon 5 of Brd4 gene. Immunoprecipitation (IP) of Brd4, RbBP5, and CDK9 was done in nuclear extracts prepared at D2 of adipogenesis. The immunoprecipitates were analyzed by western blot using antibodies indicated on the right. The asterisk (*) indicates IgG heavy chain

were enriched with motifs of adipogenic TFs such as C/EBPα, C/EBPβ, and PPARγ as well as ATF4, which was recently identified as a novel TF that promotes adipogenesis[22]. ChIP-Seq analyses of C/EBPα, C/EBPβ, and PPARγ at D2 of adipogenesis confirmed the genomic co-occupancy of Brd4 with these adipogenic TFs (Fig. 4b, c). Particularly, over 50% of the Brd4 binding regions showed co-occupancy with C/EBPα/β. Consistent with previous reports that Brd4 interacts with C/EBPα/β[11,21], we observed a physical interaction between Brd4 and C/EBPβ during adipogenesis (Fig. 4d). In C2C12 myocytes, Brd4 binding regions were enriched with motifs of myogenic TF MyoD and its binding partner TCF3 (Supplementary Fig 5a)[23]. We also confirmed genomic co-localization of Brd4 with MyoD in C2C12 myocytes (Supplementary Fig. 5b, c).

Next, we characterized the genomic features of Brd4 binding regions. Based on histone modifications, four types of regulatory elements were defined as described[19]: active enhancer, primed enhancer (previously described as silent enhancer), active promoter, and silent promoter (Fig. 5a). Interestingly, Brd4 binding sites were mainly located on active enhancers at D0 and D2 but on active promoters at D7 (Fig. 5b). Consistently, Brd4

co-localized with adipogenic TFs on active enhancers at D2 (Supplementary Fig 6). Adipogenic enhancers were defined as active enhancers that are bound by C/EBPs or PPARγ[19]. Consistent with the genomic distribution, Brd4 was highly enriched on adipogenic enhancers at D2 but enriched on adipogenic promoters, which associate with adipogenic enhancers, at D7 (Fig. 5c). At D2, Brd4 binding was observed on 43.9%, 41.7%, and 78.8% of C/EBP+PPARγ−, C/EBP−PPARγ+, and C/EBP+PPARγ+ adipogenic enhancers, respectively (Fig. 5d). Together, these results indicate that Brd4 co-localizes with LDTFs on active enhancers during adipogenesis.

**LDTFs, MLL3/MLL4, and p300 recruit Brd4 to active enhancers**. Next, we investigated the mechanisms that recruit Brd4 to active enhancers during adipogenesis. Notably, 73.8% (6129/8309) of Brd4 binding sites within active enhancers were occupied by enhancer epigenomic writers MLL4 and/or p300, indicating a substantial overlap (Fig. 6a). Among the 8309 Brd4+ active enhancers, substantially more of them were co-occupied by both epigenomic writers (p300 or MLL4) and LDTFs (C/EBPs or PPARγ) than by either alone (Fig. 6b, c).

**a**

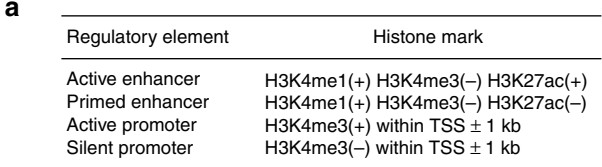

| Regulatory element | Histone mark |
|---|---|
| Active enhancer | H3K4me1(+) H3K4me3(−) H3K27ac(+) |
| Primed enhancer | H3K4me1(+) H3K4me3(−) H3K27ac(−) |
| Active promoter | H3K4me3(+) within TSS ± 1 kb |
| Silent promoter | H3K4me3(−) within TSS ± 1 kb |

**b** Brd4 genomic distribution

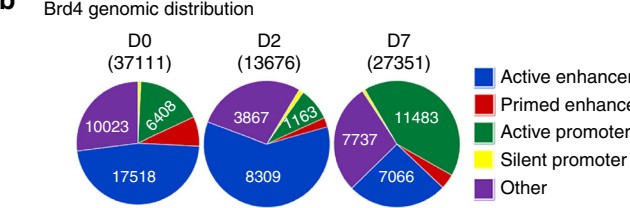

**c** Average profile of Brd4

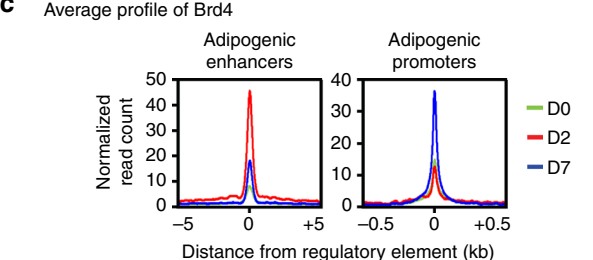

**d** Heat maps (D2 adipogenic enhancers)

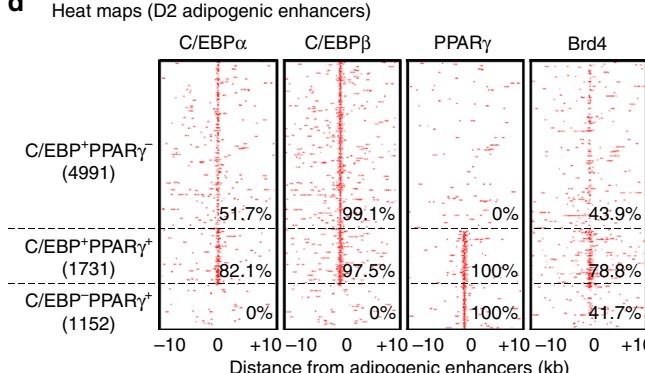

**Fig. 5** Brd4 co-localizes with LDTFs on active enhancers during adipogenesis. **a** Definition of regulatory elements by histone marks. **b** Pie charts depicting the genomic distributions of Brd4 binding regions during adipogenesis. **c** Average binding profiles of Brd4 around the center of adipogenic enhancers and promoters. Adipogenic enhancers were defined as active enhancers bound with C/EBPα, C/EBPβ, or PPARγ at D2 of adipogenesis as described previously[19]. Adipogenic promoters were defined as promoters that are closest to adipogenic enhancers. **d** Brd4 co-localizes with LDTFs on active enhancers during adipogenesis. The binding profiles of C/EBPα, C/EBPβ, PPARγ, and Brd4 on the three types of adipogenic enhancers (C/EBP⁺PPARγ⁻, C/EBP⁻PPARγ⁺, and C/EBP⁺PPARγ⁺) are shown in heat maps

Since MLL3/MLL4 facilitate CBP/p300 binding to enhancers during cell differentiation[24,25] and CBP/p300 facilitate Brd4 binding to active enhancers[11], we hypothesized that MLL3/MLL4 are required for Brd4 recruitment to enhancers during differentiation. To test this hypothesis, we performed Brd4 ChIP-Seq in *Mll3/Mll4* double KO cells at D2 of adipogenesis. Although deletion of *Mll3/Mll4* did not affect Brd4 protein levels (Fig. 6d), it led to the loss of 90.3% of total Brd4 binding sites at D2 (Fig. 6e). Consistently, we observed a marked decrease in Brd4 binding levels on MLL4⁺ active enhancers in *Mll3/Mll4* KO cells

(Fig. 6f). Decreased binding of Brd4 in *Mll3/Mll4* KO cells could be due to reduced binding of LDTFs to enhancers. Indeed, 60% (3725/6158) of C/EBPβ⁺ MLL4⁺ active enhancers showed decreased C/EBPβ binding in *Mll3/Mll4* KO cells, which consequently led to decreases in p300 and Brd4 binding (Fig. 6g). However, on the remaining 40% (2433/6158) of C/EBPβ⁺ MLL4⁺ active enhancers, deletion of *Mll3/Mll4* in preadipocytes did not affect C/EBPβ binding but prevented p300 and Brd4 binding at D2 of adipogenesis. Deletion of *Mll3/Mll4* also reduced p300 and Brd4 binding to enhancers on *Pparg* gene locus during adipogenesis (Fig. 6h). Together, these results suggest that LDTFs and epigenomic writers MLL3/MLL4 and p300 coordinate to recruit Brd4 to active enhancers during adipogenesis.

**Brd4 is required for Pol II binding on active enhancers**. We next asked how Brd4 regulates cell identity gene induction during adipogenesis. For this purpose, we selected several adipogenic enhancers (e1–e8) on cell identity genes *Pparg*, *Cebpa*, and *Fabp4*, which are bound by Brd4 during (D2) adipogenesis (Fig. 7a). We examined the occupancy of the early adipogenic TF C/EBPβ, MLL4, MLL3/MLL4-mediated H3K4me1, CBP/p300-mediated H3K27ac, Brd4, the MED1 subunit of the Mediator coactivator complex, the TBP subunit of the general transcription factor (GTF) TFIID[26], Pol II, and the catalytic subunit CDK9 of the positive transcription elongation factor b (p-TEFb)[27] on these enhancers during adipogenesis. We did not observe changes in C/EBPβ and MLL4 binding as well as H3K4me1 and H3K27ac levels on Brd4⁺ adipogenic enhancers in *Brd4* KO cells. However, deletion of *Brd4* markedly reduced MED1, TBP, Pol II, and CDK9 binding on Brd4⁺ adipogenic enhancers but not on enhancers near constitutively active genes *Arid1a* and *Jak1* (n1 and n2) (Fig. 7b). Accordingly, deletion of *Brd4* decreased eRNA production from Brd4⁺ adipogenic enhancers (Fig. 7c). These data suggest a model that sequential binding of LDTFs, epigenomic writers MLL3/MLL4, and CBP/p300 facilitates Brd4 binding on active enhancers, which is required for enhancer binding of Mediator, TFIID, Pol II and p-TEFb, eRNA production, and cell identity gene induction during adipogenesis (Fig. 7d).

**Brd4 is largely dispensable for maintaining adipocytes**. Terminally differentiated adipocytes express high levels of cell identity genes including master adipogenic TFs *Pparg* and *Cebpa*. We asked whether Brd4 is required to maintain adipocyte gene expression. To this end, we crossed *Brd4*ᶠ/ᶠ mice with *Adipoq-Cre* mice to generate adipocyte-specific *Brd4* KO mice[28]. Deletion of *Brd4* was successful in adipose tissues (Fig. 8a, b). However, we did not observe any discernable differences in adipose tissue mass (Fig. 8c, d) or the expression of adipocyte identity genes *Pparg*, *Cebpa*, *Fabp4*, or BAT marker gene *Ucp1* (Fig. 8e). RNA-Seq analysis of BAT and epididymal white adipose tissue (eWAT) from *Brd4*ᶠ/ᶠ;*Adipoq-Cre* mice confirmed that the deletion of *Brd4* in adipose tissues does not affect adipocyte and BAT-enriched gene expression (Fig. 8f, g). Our data suggest that while Brd4 is essential for adipose tissue development, it is largely dispensable for the maintenance of adipose tissues and related gene expression.

**JQ1 blocks PPARγ-stimulated adipogenesis**. Since Brd4 is essential for the induction of PPARγ in the early stage of adipogenesis, we tested whether forced expression of PPARγ could rescue adipogenesis in *Brd4* KO cells. Indeed, adipogenesis defects in *Brd4* KO cells could be rescued by retroviral vector-mediated expression of ectopic PPARγ (Fig. 9a, b). Interestingly, the expression levels of *Brd2* and *Brd3* increased in *Brd4* KO cells,

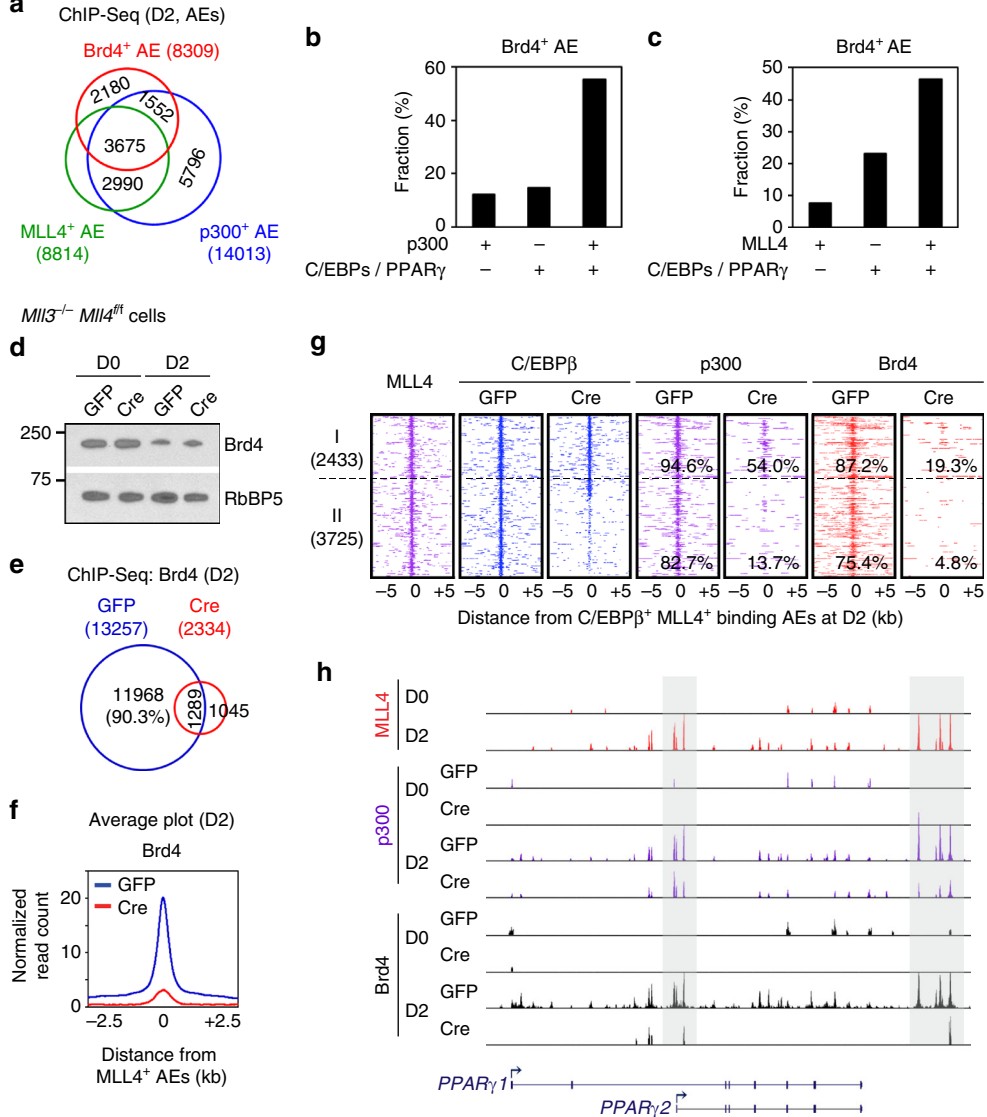

**Fig. 6** LDTFs, MLL3/MLL4, and p300 coordinate to recruit Brd4 to active enhancers. **a** Venn diagram of co-localization of Brd4, MLL4, and p300 on active enhancers (AEs) at D2 of adipogenesis. **b**, **c** Brd4 binding is highly enriched on AEs that are bound by both LDTFs (C/EBPs or PPARγ) and p300 **b** or MLL4 **c** during adipogenesis. **d**–**h** MLL3/MLL4 are required for Brd4 binding on AEs during adipogenesis. Immortalized $Mll3^{-/-}$ $Mll4^{f/f}$ preadipocytes were infected with adenoviral GFP or Cre as described previously[19], followed by adipogenesis assay. Cells were collected before (D0) and during (D2) adipogenesis for ChIP-Seq of Brd4. **d** Deletion of MLL3/MLL4 does not affect Brd4 protein levels. **e** Venn diagram representing Brd4 binding regions in GFP- and Cre-infected $Mll3^{-/-}$ $Mll4^{f/f}$ cells at D2 of adipogenesis. **f** Average profile of Brd4 binding on MLL4+ AEs at D2. **g** Heat maps around C/EBPβ+ MLL4+ AEs at D2. AEs were classified into two groups: unchanged (group I) and decreased (group II) C/EBPβ binding in $Mll3/Mll4$ double KO cells. **h** Genome browser view of p300 and Brd4 binding on $Pparg$ locus in GFP- and Cre-infected $Mll3^{-/-}$ $Mll4^{f/f}$ cells

suggesting a functional redundancy among BET family proteins (Fig. 9b). Inhibiting BET proteins by JQ1 treatment completely blocked PPARγ-stimulated adipogenesis (Fig. 9a, b). These results indicate that Brd4 is the major functional BET protein before the induction of PPARγ and suggest that Brd4 is functionally redundant with Brd2/Brd3 in promoting adipocyte gene expression downstream of PPARγ. Consistent with these results, inhibiting BET proteins by JQ1 treatment inhibited the synthetic PPARγ ligand rosiglitazone (Rosi)-induced expression of PPARγ target genes *Cebpa*, *Adipoq*, and *Fabp4* in undifferentiated pre-adipocytes expressing ectopic PPARγ (Fig. 7c), indicating that BET family proteins are required for PPARγ target gene expression.

## Discussion

Using two independently developed *Brd4* conditional KO mice and derived cell lines, we demonstrate that Brd4 is essential for adipogenesis and myogenesis in culture and in vivo. Using RNA-Seq, we show Brd4 promotes cell-type-specific gene expression during cell differentiation. Using ChIP-Seq, we show Brd4 predominantly binds to active enhancers during adipogenesis but preferentially on active promoters after adipogenesis. Further, enhancer epigenomic writers MLL3/MLL4 are required for Brd4 binding to active enhancers during adipogenesis. Finally, Brd4 facilitates enhancer binding of Mediator, TFIID, Pol II, and p-TEFb, and eRNA transcription during adipogenesis. Our findings identify Brd4 as an enhancer epigenomic reader that connects active enhancers with cell identity gene induction during

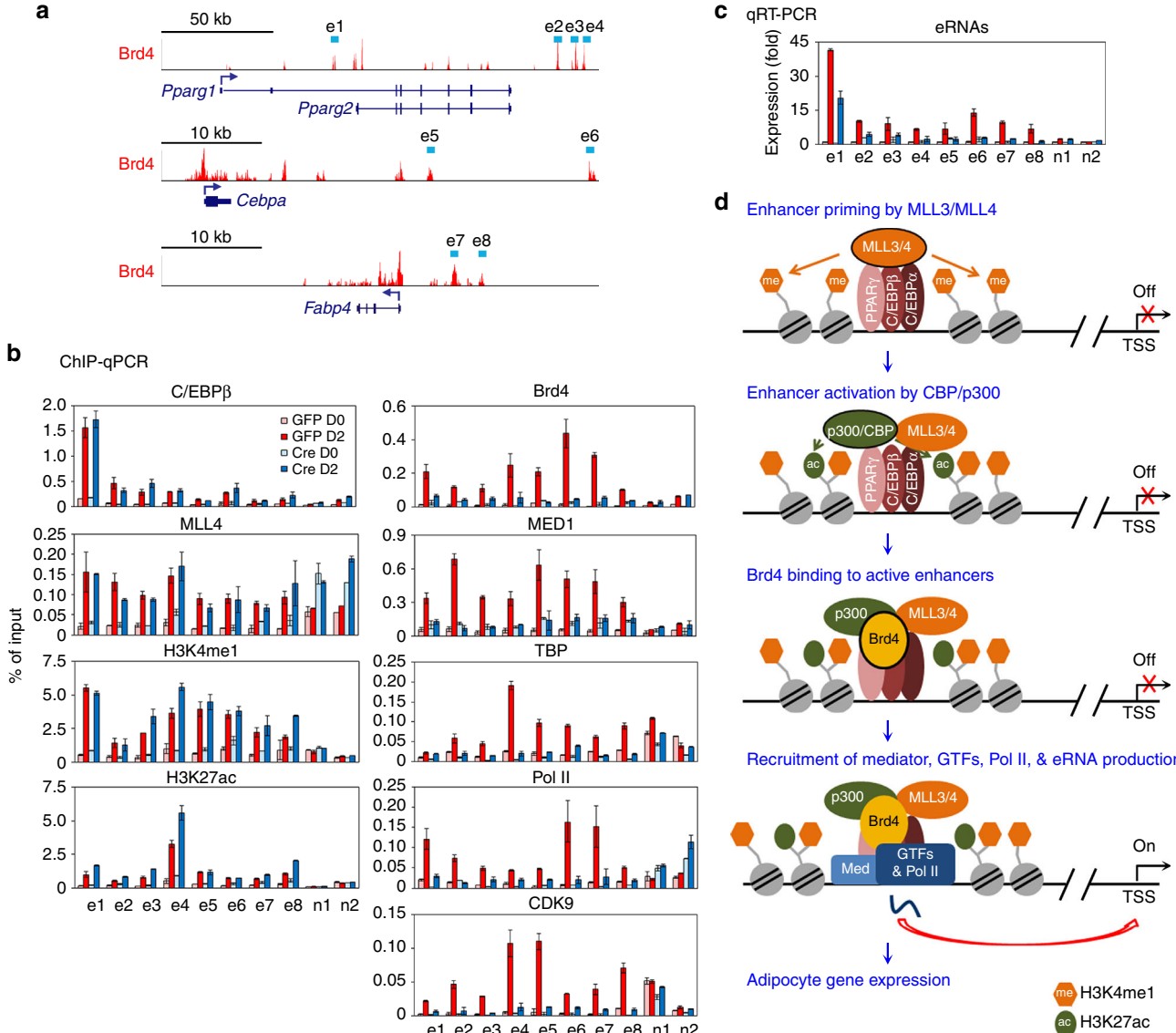

**Fig. 7** Brd4 is required for Pol II binding on active enhancers during adipogenesis. **a** Schematic of genomic locations of Brd4+ enhancers (e1–e8) on *Pparg*, *Cebpa*, and *Fabp4* loci. **b** Brd4 is dispensable for C/EBPβ, MLL4, H3K4me1, and H3K27ac enrichment, but is essential for MED1, TBP, Pol II, and CDK9 binding, on adipogenic enhancers. ChIP-qPCR analyses of indicated factors are shown on adipogenic enhancers (e1–e8). Enhancers of constitutively expressed genes *Arid1a* (n1) and *Jak1* (n2) were chosen as negative control regions. **c** Brd4 is required for eRNA transcription on adipogenic enhancers. The data are presented as means ± SD. Three technical replicates from a single experiment were used. **d** Model depicting that sequential actions of LDTFs, H3K4 mono-methyltransferases MLL3/MLL4, H3K27 acetyltransferases CBP/p300, epigenomic reader Brd4, transcription coactivator Mediator, and Pol II on enhancers control cell identity gene expression during differentiation

differentiation. Together with previous findings in the literature, our data suggest a model in which sequential actions of LDTFs, H3K4 mono-methyltransferases MLL3/MLL4, H3K27 acetyl-transferases CBP/p300, epigenomic reader Brd4, transcription coactivator Mediator, and Pol II transcription machinery on enhancers control cell identity gene expression during differentiation.

Brd4 inhibitors are promising drug candidates for treating cancers and other diseases[29,30]. However, few studies have looked at the role of Brd4 in cell differentiation and animal development, except that Brd4 is required for the differentiation of erythroid and osteoblast in cell culture[31,32], that inducible knockdown of *Brd4* in mice results in skin hyperplasia and loss of cell diversity in intestine[33], and that myeloid lineage-specific deletion of *Brd4*

leads to the compromised innate immune response[34]. By crossing *Brd4*f/f with Myf5-Cre mice, we demonstrate that Brd4 is required for adipose tissue and muscle development in vivo. By knocking down *Brd4* in 3T3-L1 white preadipocytes and C2C12 myoblasts and knocking out *Brd4* in brown preadipocytes, we confirmed that the essential role of Brd4 in adipogenesis and myogenesis is cell-autonomous. Our finding on Brd4 in adipogenesis is consistent with a previous report that JQ1 inhibits differentiation of mesenchymal C3H10T1/2 cells toward adipocytes in culture[35]. JQ1 treatment cannot distinguish functional roles of BET proteins in adipogenesis. Our genetic study distinguishes the roles of BET family members in adipogenesis and indicates that Brd4 is the major BET protein controlling PPARγ induction in the early phase of adipogenesis while the functionally redundant Brd2,

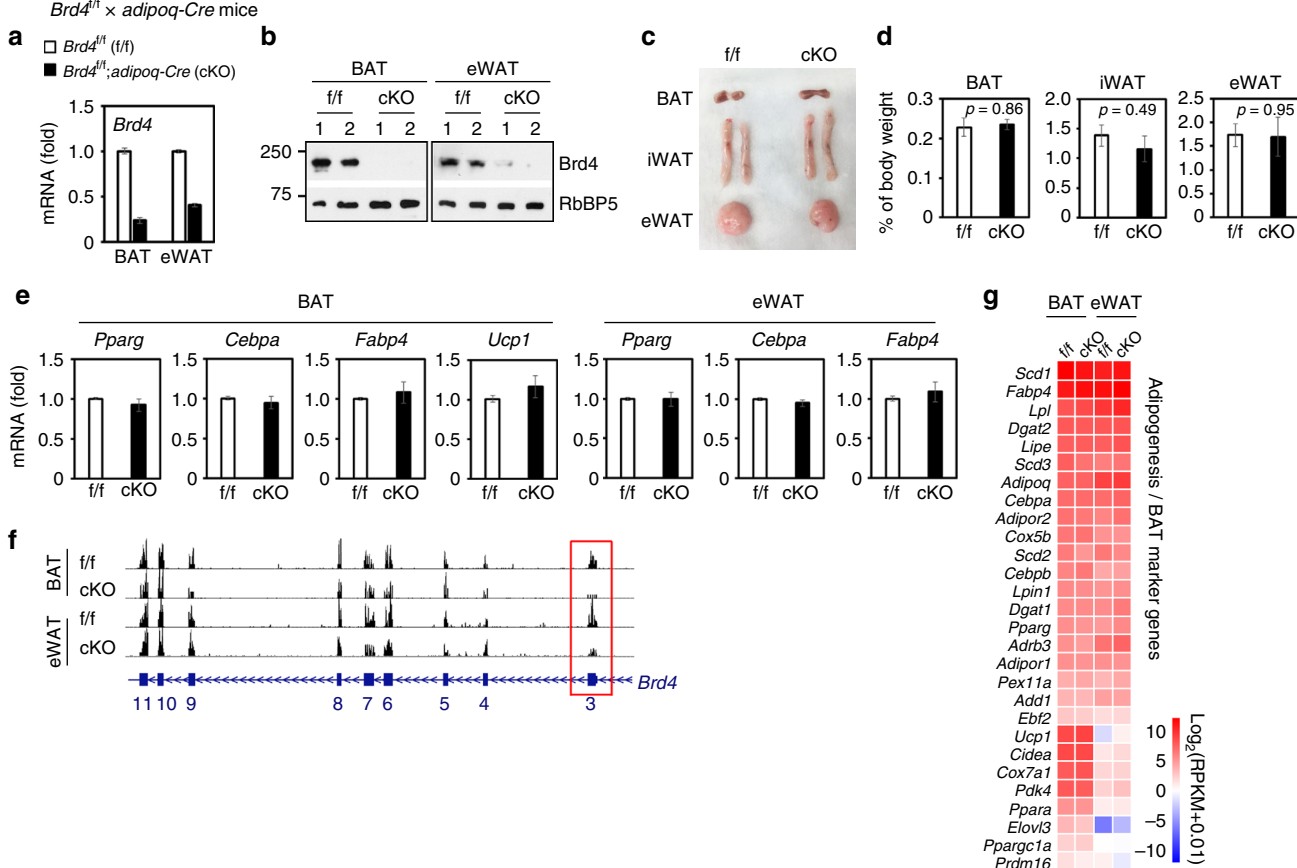

**Fig. 8** Brd4 is dispensable for the maintenance of cell identity gene expression in adipocytes. *Brd4*[f/f] mice were crossed with *Adipoq-Cre* to generate adipocyte-specific *Brd4* KO mice. **a** qRT-PCR confirmation of Adipoq-Cre-mediated *Brd4* deletion in mouse adipose tissues (*n* = 6 per group). **b** Western blot analysis of Brd4 in BAT and eWAT. **c** Representative morphology of mouse adipose tissues. **d** Adipose tissue mass was normalized to body weight (*n* = 6 per group). Statistical comparison between groups was performed using Student's *t* -test. The data are presented as means ± SEM. **e** qRT-PCR of *Pparg, Cebpa, Fabp4,* and *Ucp1* expression in adipose tissues (*n* = 4 per group). Quantitative PCR data in this figure are presented as means ± SEM. **f, g** Deletion of Brd4 does not affect adipocyte gene expression in adipose tissues. Equal amounts of total RNA from BAT and eWAT of four 12-week-old male mice were combined for RNA-Seq analysis. **f** Genome browser view of RNA-Seq analysis on *Brd4* gene locus. The targeted exon 3 is highlighted in red box. **g** Heat map of RNA-Seq values of adipogenesis/BAT marker genes

Brd3, and Brd4 control the induction of PPARγ downstream adipocyte genes.

Brd4 localizes on both active promoters and active enhancers in human and mouse tumor cells[10,11]. In this study, we show dynamic changes of Brd4 binding regions at various stages of differentiation. In our adipogenesis model system, Brd4 mainly localizes on active enhancers that are associated with cell identity genes induced during differentiation. After differentiation, Brd4 binding on active enhancers largely remains but with reduced binding intensity. Interestingly, Brd4 binds to ~75% of all active promoters in differentiated cells, while only ~8% of all active promoters are occupied by Brd4 during differentiation. These results suggest distinct functions of Brd4 during and after cell differentiation. In differentiating cells, Brd4 localization on active enhancers controls cell identity gene induction necessary for terminal differentiation. The role of Brd4 on active promoters in terminally differentiated cells remains to be understood.

LDTFs bind to cell-type-specific enhancers and recruit H3K4 mono-methyltransferases MLL3/MLL4[19]. MLL3/MLL4 are required for enhancer binding of H3K27 acetyltransferases CBP/p300 and enhancer activation during adipogenesis and embryonic

stem cell differentiation[24,25]. It has been shown that hematopoietic LDTFs and CBP/p300 facilitate Brd4 recruitment to active enhancers in leukemia cells[11]. Similarly, our data show that adipogenic TFs and p300 cooperatively recruit Brd4 to active enhancers during adipogenesis. Furthermore, we demonstrate that MLL3/MLL4 are critical for Brd4 binding on enhancers. Consistent with our data, a recent study showed that Brd4 is required for binding of Mediator and CDK9 on enhancers including the *Myc* super-enhancer in mouse acute myeloid leukemia (AML cells)[13]. Brd4 may regulate Mediator binding to enhancers through its physical interaction with Mediator complex[27]. On the other hand, since PPARγ physically associates with Mediator complex as well[36], it is also possible that decreased Mediator binding in Brd4 KO cells is due to the decreased expression of PPARγ, a direct target of Brd4. Physical and functional association between Brd4 and CDK9, a component of p-TEFb, has been well documented[27,37]. Our data suggest that Brd4 is a molecular bridge between cell-type-specific enhancers and general transcription machinery. Taken together, we propose a model that adipocyte gene expression is induced by sequential actions of LDTFs C/EBPα, C/EBPβ, and PPARγ, enhancer

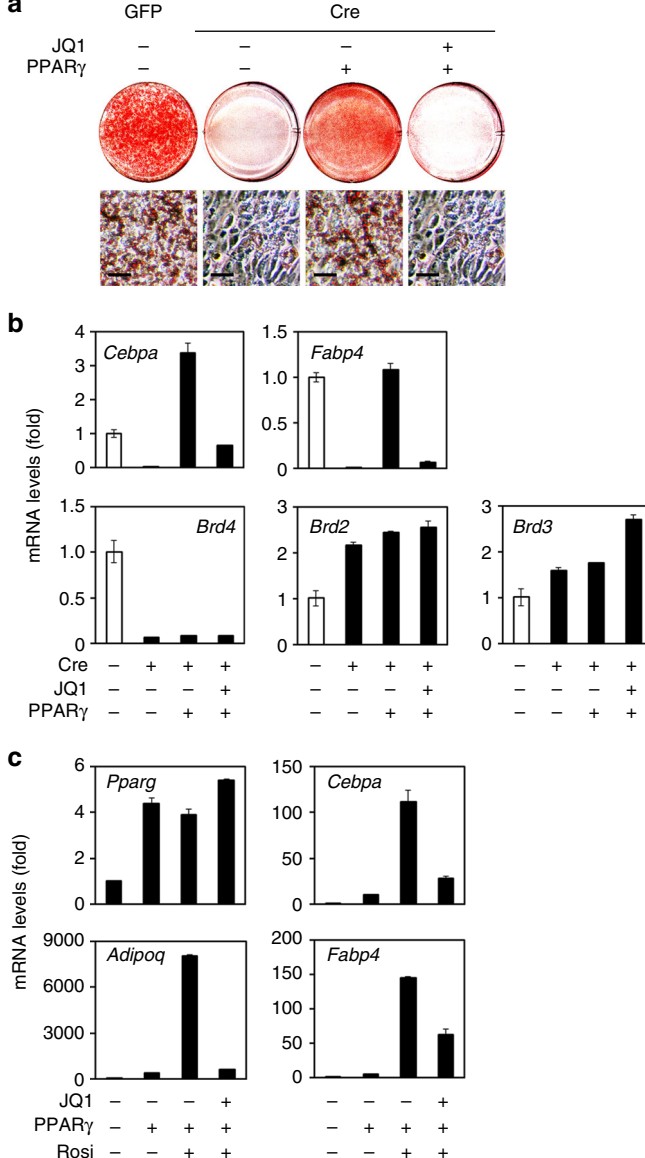

**Fig. 9** JQ1, but not the single KO of Brd4, blocks PPARγ-stimulated adipogenesis. **a**, **b** *Brd4*^f/f brown preadipocytes were infected with adenoviral GFP or Cre. Cre-infected cells were further infected with retroviruses expressing vector alone or PPARγ. After hygromycin selection, cells were induced for adipogenesis assay in the presence or absence of 0.5 μM BET inhibitor JQ1. **a** Oil red O staining at D7 of adipogenesis. Scale bars = 15 μm. **b** qRT-PCR of *Cebpa*, *Fabp4*, *Brd4*, *Brd2*, and *Brd3* at D7 of adipogenesis. **c** BET family proteins are required for ligand-induced PPARγ target gene expression. Subconfluent *Brd4*^f/f preadipocytes expressing ectopic PPARγ were pretreated with JQ1 (0.5 μM) for 3 h followed by treatment with 1 μM rosiglitazone (Rosi) for 24 h in the presence of JQ1. The data are presented as means ± SD. Three technical replicates from a single experiment were used

epigenomic writers MLL3/MLL4 and CBP/p300, enhancer epigenomic reader Brd4, Mediator, and Pol II transcription machinery on enhancers (Fig. 7d).

## Methods
**Plasmids and antibodies**. The lentiviral shRNA plasmids pLKO.1 targeting mouse *Brd4* (clone IDs TRCN0000311976, TRCN0000088480, and TRCN0000088481)

were purchased from Sigma. Anti-RbBP5 (A300–109A), anti-Brd4 (A301–985A100), and anti-MED1 (A300–793A) were from Bethyl Laboratories. Anti-C/EBPα (sc-61X), anti-C/EBPβ (sc-150X), anti-PPARγ (sc-7196X), and anti-p300 (sc-585X) were from Santa Cruz Biotechnology. Anti-H3K4me1 (ab8895), anti-H3K4me2 (ab7766), and anti-H3K27ac (ab4729) were from Abcam. Anti-Pol II (17-672) was from Millipore. For western blot analysis, all antibodies were diluted to 1 μg ml⁻¹. Uncropped blots are available in the Supplementary Figure 7.

**Generation of mouse strains**. Two *Brd4* conditional KO mouse lines were used in this study. In the first *Brd4* conditional KO mouse line (*Brd4*^f/f), the first coding exon of *Brd4* gene, exon 3, is flanked by loxP sites. Genotyping the *Brd4* alleles, PCR was done using the following primers: 5′-GCCTA-GATCAGTGCCTCCATTG-3′ and 5′-ACTGGAACTACATGGCAGCCTG-3′. PCR amplified 244 bp from the wild-type and 344 bp from the floxed allele. *Brd4*^f/f mice were crossed with *Myf5-Cre* (Jackson no. 007893, C57BL/6 J and 129S4/SvJaeSor mixed background) or *Adipoq-Cre* (C57BL/6 background)[28] to generate *Brd4*^f/f;*Myf5-Cre* or *Brd4*^f/f;*Adipoq-Cre* mice. For characterization of *Brd4*^f/f;*Adipoq-Cre* mice, we used six 12-week-old male mice per genotype. Animals were not randomized and the researchers were not blinded during the experiment and when assessing the outcome. No animals were excluded from the analysis.

To establish the second *Brd4* conditional KO mouse line (*Brd4*^f/f #2), the heterozygous conditional *Brd4* gene trap mice (*Brd4*^tm1a(EUCOMM)Wtsi) were obtained from the KOMP2 program at Baylor College of Medicine and crossed with FLP1 mice (Jackson no. 003946) to delete the neomycin cassette. In the resulting *Brd4*^f/f #2 mice, the exon 5 of *Brd4* gene is flanked by loxP sites. Genotyping of the Brd4 alleles was done using the following primers: 5′-GGACATGGTGACAGAG TGG-3′ and 5′-TCAAATGAATTCACTAGAACTAC-3′. PCR amplified 168 bp from the wild-type and 284 bp from the floxed allele. *Brd4*^f/f #2 mice were crossed with *Cre-ER* (Jackson no. 008463) to generate *Brd4*^f/f #2;*Cre-ER* mice.

All mouse experiments were performed in accordance with the NIH Guide for the Care and Use of Laboratory Animals and approved by the Animal Care and Use Committee of NIDDK, NIH.

**Histology and immunohistochemistry**. E18.5 embryos were isolated by Cesarean section, fixed in 4% paraformaldehyde, dehydrated in a methanol series, and embedded in paraffin for sectioning. Paraffin sections were stained with routine H&E or subjected to immunohistochemistry using anti-Myosin (MF20; Developmental Studies Hybridoma Bank, 1:20 dilution) and anti-UCP1 (ab10983; Abcam, 1:400 dilution) antibodies[19].

**Cell culture and differentiation assays**. Primary brown preadipocytes were isolated from interscapular BAT of newborn *Brd4*^f/f, *Brd4*^f/f #2;*Cre-ER* and immortalized by infecting retroviruses expressing SV40T[38]. Adipogenesis of immortalized brown preadipocytes was induced with DMEM supplemented with 10% fetal bovine serum (FBS), 0.02 μM insulin, 1 nM T3, 0.5 mM IBMX, 2 μg ml⁻¹ DEX, and 0.125 mM indomethacin for 2 days[39]. After this period, the culture medium was supplemented with FBS, insulin, and T3 only. 3T3-L1 cells were from Daniel Lane.

C2C12 myoblasts were purchased from ATCC and cultured in growth medium of DMEM containing 15% FBS. Myogenesis was induced by replacing growth medium to DMEM containing 2% horse serum when cells were ~70% confluent. Before changing the medium, cells were washed with plain DMEM twice.

**qRT-PCR**. Total RNA was extracted using TRIzol (Invitrogen) and reverse transcribed using ProtoScript II first-strand cDNA synthesis kit (NEB), following manufacturer's instructions. qRT-PCR of *Brd4* exon 3 was done using SYBR green primers: forward 5′-CCCAGAGACCTCCAACCCTAA-3′ and reverse 5′-AACTGGTGTTTCCATAGTGTCTTGAG-3′. qRT-PCR of Brd4 exon 5 was done using the primers: forward 5′-TGACATCGTCTTAATGGCAGAAG-3′ and reverse 5′-CCTTTTGCCTGGACTATCATGAT-3′.

**ChIP-Seq and RNA-Seq**. For ChIP-Seq analysis, formaldehyde was added directly to cell culture medium to a final concentration of 2%. After 10 min of incubation at room temperature, glycine was added to a concentration of 125 mM to quench crosslinking reaction. Approximately 2 × 10⁷ cells were washed with 20 ml cold PBS in culture dish twice and scraped off in 10 ml Farnham lysis buffer (5 mM PIPES, pH 8.0, 85 mM KCl, 0.5% NP-40, supplemented with protease inhibitors), then pelleted by centrifugation at 3000×*g* for 5 min at 4 °C. Cell pellet was resuspended in 10 ml lysis buffer and pelleted again to remove cytosolic proteins. Resulting nuclear pellet was sonicated in 2 ml TE buffer (10 mM Tris-Cl, pH 8.0, 1 mM EDTA, supplemented with protease inhibitors) to achieve DNA fragments of 200–500 bp. Detergents were added to digested chromatin fractions to make 1× RIPA buffer (10 mM Tris-Cl, pH 7.6, 1 mM EDTA, 0.1% SDS, 0.1% sodium deoxycholate, 1% triton X-100). After centrifugation, supernatant was collected in a new tube. For each ChIP, 8–10 μg antibodies were pre-incubated with 50 μl Dynabeads Protein A (Life Technologies) in 1 ml PBS overnight at 4 °C under

gentle rotation. Next day, 1 ml chromatin from $1 \times 10^7$ cells was mixed with antibody-beads complex and incubated overnight at 4 °C with gentle rotation. Chromatin immunoprecipitates were washed twice with 1 ml RIPA buffer, twice with 1 ml RIPA containing 300 mM NaCl, twice with 1 ml LiCl buffer (50 mM Tris-Cl, pH 7.5, 250 mM LiCl, 0.5% NP-40, 0.5% sodium deoxycholate), and twice with 1 ml TE buffer. Samples were reverse-crosslinked in 100 μl elution buffer (1% SDS, 0.1 M NaHCO3, and 100 μg proteinase K) overnight at 65 °C. DNA was purified by QIAquick PCR Purification Kit (QIAGEN) and quantified using Qubit dsDNA HS Assay Kit (Thermo Fisher Scientific)[19,24].

For RNA-Seq, mRNAs were purified using Dynabeads mRNA Purification Kit (Invitrogen), and then they were used to synthesize double-stranded cDNAs using SuperScript Double-Stranded cDNA Synthesis Kit (Invitrogen). Sequencing libraries were constructed using NEBNext Ultra II DNA Library Prep Kit for Illumina (NEB). All ChIP-Seq and RNA-Seq samples were sequenced on the Illumina HiSeq 2500.

**Computational analysis**. To identify Brd4 binding regions, we used 'SICER' method with a window size of 50 bp and a gap size of 50 bp[40]. To eliminate non-specific binding of Brd4 antibody, we compared Brd4 ChIP-Seq data from *Brd4* KO cells with that from two biological replicates of control (Ppar$\gamma$^f/f) cells[24], and kept only the identified Brd4 binding regions with enrichment level significantly higher in control cells than in the *Brd4* KO cells, with an estimated false discovery rate (FDR) of <$10^{-3}$. Then, we chose only the overlapping Brd4 binding regions from two biological replicates. For Brd4 ChIP-Seq data in *Mll3*−/− *Mll4*^f/f cells, an FDR of <$10^{-10}$ was used to find the high-confidence ChIP-enriched regions. Other published ChIP-Seq data sets were downloaded (GSE74189, GSE50466, and GSE44824)[19,24,41].

For motif analysis of Brd4 binding regions (Fig. 4a), we used SeqPos motif tool in Galaxy Cistrome (http://cistrome.org/ap/root)[42]. The algorithm used in this motif tool is previously described in detail[43]. We selected the top 3000 Brd4 binding regions to screen enriched TF motifs at each time point.

**Data availability**. All data sets described in the paper have been deposited in NCBI Gene Expression Omnibus under accession number GSE99101.

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

## Acknowledgements

We thank John Seavitt and the KOMP2 program at Baylor College of Medicine for Brd4^tm1a(EUCOMM)Wtsi mice, Philipp Scherer for Adipoq-Cre mice, Erin Koh and Jaena

Taitague for technical support, and NIDDK Genomics Core for sequencing. This work was supported by the Intramural Research Program of NIDDK, NIH to KG.

## Author contributions

J-E.L. and K.G. conceived and designed the experiments. J-E.L., Y-K.P., S.P., and Y.J. performed the experiments. J-E.L., Y-K.P., S.P., Y.J., N.W., B.L., W.P., and K.G. analyzed the data. A.D. and K.O. generated $Brd4^{f/f}$ mice. J-E.L., N.W., B.L., and W.P. performed computational analyses. J-E.L., S.P., and K.G. wrote the manuscript. K.G. supervised all the experiments.

## Additional information

**Competing interests:** The authors declare that they have no competing financial interests.

