## [Peer Review File · Nature Communications]

Reviewers' comments:

Reviewer #1 (Remarks to the Author):

The manuscript by Lee et al, reveals fundamental insights into enhancer-mediated gene regulation by dissecting the order of events upstream and downstream of the protein BRD4. Overall, I was impressed by this work. The experiments are rigorous, particularly the use of flox/Cre alleles of BRD4 to study mechanisms, as well as the evaluation of each step of enhancer activation- from TF binding to recruitment of each coactivator component (MLL3/p300/BRD4/Mediator). The use of adipocyte differentiation was also highly effective in this study, and leads to conclusions with potential relevance to normal physiology and pathologies of this cell lineage. Finally, I found the upstream role of MLL3/MLL4 in facilitating BRD4 recruitment to be of major significance to the field of enhancer biology. I strongly endorse publication of this study in Nature Communications without delay.

Reviewer #2 (Remarks to the Author):

Ge and his colleagues utilized tissue-specific Brd4 KO mice to investigate its roles in adipogenesis and myogenesis. They have comprehensively described the distinctive roles of Brd4 before and after cellular differentiation, demonstrating that Brd4 is essentially important for adipogenesis and myogenesis, but largely dispensable in differentiated cells. This is an important characterization of an important epigenetic regulator in cell fate determination. The manuscript is clearly written with robust data presentation, though this reviewer has a few minor comments here.

1. The Brd4 expression level was dramatically decrease along the course of cellular differentiation (Fig 3B), do the authors have any thoughts on this? Also the sharply decreased Brd4 seemed not to be reflected by the ChIP-seq data, because the Brd4 binding peaks appeared to increase in the differentiated cells. This should be explained and explored.

2. Interestingly, in the literature, the expression level of Mediator components also decreased during myogenesis. Probably all the transcriptional machinery are going down as well. One wonders if the total transcribed mRNA will also go down during the differentiation.

3. The authors tried to come up with an unified model of sequential order of binding/modification of LDTFs, MLL-H3K4me1, CBP/P300-H3K27ac, Brd4, Mediator, and Pol II machinery, which is convincing. However, did the authors noticed any cell type-specific order of events or any difference between the adipogenesis and myogenesis, besides the LDTFs?

4. Brd4 seems to be required for recruitment of Mediator complex, could the authors offer any mechanistic insight on this based on their ChIP-seq data analysis in this study or from the literature?

Reviewer #3 (Remarks to the Author):

Lee and colleagues investigate the role of BRD4 in adipogenesis and myogenesis using conditional knockout mice. They reveal that genetic ablation of Brd4 in Myf5+ cells impairs muscle development, disrupts brown adipose tissue formation, and leads to perinatal lethality. In contrast, use of adiponectin-CRE to remove Brd4 in later stages of adipogenesis shows no clear phenotype. Genome-wide profiling of BRD4 in wild type and MLL3/MLL4-deficient cells reveals genome-wide occupancy that is dependent on MLL3/4 activity. Moreover, BRD4-deficiency is correlated with loss

of MED1 and RNAPII occupancy at adipogenic enhancers, supporting a model by which BRD4 is a necessary molecular bridge between H3K27 acetylation and recruitment of the general transcriptional machinery. While the study is well designed, and the data are robust and interesting, it is concerning that many of the major findings are similar to those from earlier studies of BET family proteins, including Brd4, in adipogenic and myogenic differentiation. This concern would be eliminated if the authors could provide insight into how BRD4 affects recruitment of the general transcriptional machinery. Other concerns follow.

1. Interpreting the adiponectin-CRE model for BRD4 function in mature adipocytes is difficult without a western blot for BRD4 from the adipocyte fraction of adipose tissue. Approximately 30-40% of Brd4 mRNA persists in both the BAT and eWAT of these animals as revealed in Figure 8A. While the remaining Brd4 mRNA could derive from other cell populations such as the SVF, it is also possible that the knockout of Brd4 in mature adipocytes is incomplete and sufficient levels of BRD4 remain to carry out genomic functions. The western blot would directly address this concern.
2. Related to point 1, the manuscript would be strengthened by demonstrating that the cell-autonomous function of BRD4 in pre-adipocytes is dispensable following forced expression of either PPAR γ or CEBP α to bypass the early stages of adipogenic commitment.
3. The co-IP of BRD4 and CEBP β is not compelling in Figure 4D given a lack of controls and the faint signals for BRD4 and CEBP β in the IP from CRE-treated cells. To address specificity, the authors should probe for another nuclear protein in the BRD4 IP and perform a control IP of another nuclear protein. These experiments will determine whether the association between BRD4 and CEBP β is specific or a result of general enrichment for transcription factors and/or chromatin in the IP fraction.
4. In Figure 3, the authors report changes to the BRD4 cistrome (panel A) and abundance (panel B) during adipogenesis. Can the authors provide insight into whether the changes in BRD4 occupancy result from genomic redistribution of BRD4, decreased abundance of BRD4 or both? Heatmaps showing read counts at all binding sites will be more informative than Venn diagrams that rely on hard cut-offs.

Minor concerns:

5. Figures 4A-C give essentially the same results as figures 5C-E. Some of these data can be moved to the supplemental information to streamline the paper without a loss of information.
6. More details on the motif mining approach used for Figure 4A should be included in the methods. It is unclear as to which bioinformatic tools were utilized, which genomic regions were used for the background and whether the background properly compensates for differences in nucleotide composition/motif prevalence in promoters vs enhancers. This is particularly important given that the proportion of binding sites at enhancers versus promoters changes dramatically over the adipogenic time course used to map Brd4 occupancy.
7. Scale bars should be included with the micrograph images of Figure 1D.
8. Page 5, line 90: "Consistently, knockdown of Brd4 in 3T3-L1...." The use of the word "consistently" is confusing. Does it mean that the knockdown experiment worked every time, or that the knockdown produces a phenotype consistent with the genetic knockout? This should be rephrased to communicate the correct idea unambiguously.

Reviewers' comments:

Reviewer #1 (Remarks to the Author):

The manuscript by Lee et al, reveals fundamental insights into enhancer-mediated gene regulation by dissecting the order of events upstream and downstream of the protein BRD4. Overall, I was impressed by this work. The experiments are rigorous, particularly the use of flox/Cre alleles of BRD4 to study mechanisms, as well as the evaluation of each step of enhancer activation- from TF binding to recruitment of each coactivator component (MLL3/p300/BRD4/Mediator). The use of adipocyte differentiation was also highly effective in this study, and leads to conclusions with potential relevance to normal physiology and pathologies of this cell lineage. Finally, I found the upstream role of MLL3/MLL4 in facilitating BRD4 recruitment to be of major significance to the field of enhancer biology. I strongly endorse publication of this study in Nature Communications without delay.

We greatly appreciate the Reviewer #1's favorable review of our manuscript.

Reviewer #2 (Remarks to the Author):

Ge and his colleagues utilized tissue-specific Brd4 KO mice to investigate its roles in adipogenesis and myogenesis. They have comprehensively described the distinctive roles of Brd4 before and after cellular differentiation, demonstrating that Brd4 is essentially important for adipogenesis and myogenesis, but largely dispensable in differentiated cells. This is an important characterization of an important epigenetic regulator in cell fate determination. The manuscript is clearly written with robust data presentation, though this reviewer has a few minor comments here.

1. The Brd4 expression level was dramatically decrease along the course of cellular differentiation (Fig 3B), do the authors have any thoughts on this? Also, the sharply decreased Brd4 seemed not to be reflected by the ChIP-seq data, because the Brd4 binding peaks appeared to increase in the differentiated cells. This should be explained and explored.

We thank Reviewer #2 for careful reading of our manuscript and appreciate the highly constructive comments. Since this point was raised by both Reviewers #2 and #3 (see point 4), we have reexamined Brd4 protein levels during adipogenesis using nuclear extracts. We repeated the Western blot twice using nuclear extracts from 2 different cell lines and found that Brd4 levels decrease mildly at D2 of adipogenesis. These new Western data are consistent with the qRT-PCR data of Brd4 shown in Figure 2e. We have replaced the original Western blot data obtained from whole cell lysates with the new data obtained from nuclear extracts (the new Figure 3b). In addition, we have generated heatmaps of all Brd4 binding sites during adipogenesis and included it in the new Figure 3c. This new data clearly shows the redistribution of Brd4 during adipogenesis and provides more useful information.

2. Interestingly, in the literature, the expression level of Mediator components also decreased during myogenesis. Probably all the transcriptional machinery are going down as well. One wonders if the total transcribed mRNA will also go down during the differentiation.

Expression of transcriptional machinery including Mediator components and Brd4 decreases during myogenesis¹⁻³. We found that total mRNA transcripts decrease during adipogenesis (see Figure 1 for Reviewer #2 below). Interestingly, it has been shown that total mRNA levels decrease 2-fold during differentiation of mouse embryonic stem cells to neural precursor cells⁴. Although this is a very interesting observation, we feel that it does not affect our conclusion and is beyond the scope of this

manuscript. Therefore, we summarized the data for Reviewer #2 but decided not to include it in the manuscript.

Figure 1 for Reviewer #2.
Total mRNA levels during adipogenesis

3. The authors tried to come up with an unified model of sequential order of binding/modification of LDTFs, MLL-H3K4me1, CBP/P300-H3K27ac, Brd4, Mediator, and Pol II machinery, which is convincing. However, did the authors noticed any cell type-specific order of events or any difference between the adipogenesis and myogenesis, besides the LDTFs?

The induction of PPAR γ in the early phase of adipogenesis is under the control of the pioneering LDTF C/EBP β . The expression of C/EBP β is unaltered by Brd4 depletion in adipogenesis. In contrast, MyoD expression is decreased by Brd4 knockdown in C2C12 myogenesis (Supplementary Figure 4d). Therefore, an ideal model to address this issue will be to ectopically express MyoD in C2C12 cells or fibroblasts followed by induction of myogenesis and then examine the sequential order of upstream and downstream events of Brd4 binding in myogenesis. We feel such experiments are beyond the scope of the current manuscript.

We'd like to point out that our findings in adipogenesis are highly consistent with previous findings from the Vakoc lab that Brd4 works downstream of hematopoietic transcription factors and CBP/p300 but upstream of the mediator on enhancers in acute myeloid leukemia (AML) cells ^{5,6}.

4. Brd4 seems to be required for recruitment of Mediator complex, could the authors offer any mechanistic insight on this based on their ChIP-seq data analysis in this study or from the literature?

Similar with our data, a previous study showed that Brd4 is required for Mediator binding on enhancers in mouse AML cells ⁶. We think that Brd4 may regulate Mediator binding to enhancers through its physical interaction with Mediator complex ⁷. On the other hand, decreased binding of Mediator on enhancers in Brd4 KO cells could be due to the decreased expression of PPAR γ , which is known to physically associate with the Mediator complex ⁸. Our new data suggest that Brd4 is also required for the recruitment of TBP of TFIID and CDK9 component of the p-TEFb elongation factor (updated Figure 7b). Therefore, Brd4 appears to act as a molecular bridge between cell type-specific enhancers and transcription machinery. We have updated the Discussion to reflect the new data and mechanistic insight.

Reviewer #3 (Remarks to the Author):

Lee and colleagues investigate the role of BRD4 in adipogenesis and myogenesis using conditional

knockout mice. They reveal that genetic ablation of Brd4 in Myf5+ cells impairs muscle development, disrupts brown adipose tissue formation, and leads to perinatal lethality. In contrast, use of adiponectin-CRE to remove Brd4 in later stages of adipogenesis shows no clear phenotype. Genome-wide profiling of BRD4 in wild type and MLL3/MLL4-deficient cells reveals genome-wide occupancy that is dependent on MLL3/4 activity. Moreover, BRD4-deficiency is correlated with loss of MED1 and RNAPII occupancy at adipogenic enhancers, supporting a model by which BRD4 is a necessary molecular bridge between H3K27 acetylation and recruitment of the general transcriptional machinery. While the study is well designed, and the data are robust and interesting, it is concerning that many of the major findings are similar to those from earlier studies of BET family proteins, including Brd4, in adipogenic and myogenic differentiation. This concern would be eliminated if the authors could provide insight into how BRD4 affects recruitment of the general transcriptional machinery. Other concerns follow.

We thank Reviewer #3 for careful reading of our manuscript and appreciate highly constructive comments. Following the reviewer's suggestion, we further investigated the recruitment of the TBP subunit of the general transcription factor TFIID and the CDK9 subunit of p-TEFb elongation factor on adipogenic enhancers in wild type and Brd4-deficient preadipocytes during adipogenesis. In the new Figure 7b, we show that Brd4 is required for the enhancers binding of Pol II, Mediator, TBP as well as CDK9. We have updated our model in Figure 7d and Discussion based on this new finding.

1. Interpreting the adiponectin-CRE model for BRD4 function in mature adipocytes is difficult without a western blot for BRD4 from the adipocyte fraction of adipose tissue. Approximately 30-40% of Brd4 mRNA persists in both the BAT and eWAT of these animals as revealed in Figure 8A. While the remaining Brd4 mRNA could derive from other cell populations such as the SVF, it is also possible that the knockout of Brd4 in mature adipocytes is incomplete and sufficient levels of BRD4 remain to carry out genomic functions. The western blot would directly address this concern.

We have included the Western blot of Brd4 in BAT and eWAT of *Brd4^{fl/fl};Adipoq-Cre* mice in the new Figure 8b. It appears that the efficiency of Adipoq-Cre-mediated deletion of Brd4 in mature BAT and eWAT was very good, which is consistent with the RNA-Seq data of Brd4 exon 3 (Figure 8f).

2. Related to point 1, the manuscript would be strengthened by demonstrating that the cell-autonomous function of BRD4 in pre-adipocytes is dispensable following forced expression of either PPAR γ or CEBP α to bypass the early stages of adipogenic commitment.

Following Reviewer #3's suggestion, we performed adipogenesis assay in Brd4-deficient preadipocytes ectopically expressing PPAR γ . As shown in the new Figure 9a, forced expression of PPAR γ indeed rescued adipogenesis in Brd4-deficient cells, but JQ1 treatment completely blocked PPAR γ -stimulated adipogenesis. Interestingly, expression levels of *Brd2* and *Brd3* increased in *Brd4* KO cells, suggesting a functional redundancy among BET family proteins (new Figure 9b). These data suggest that Brd4 is the major BET protein controlling the induction of PPAR γ expression while the functionally redundant BET proteins control the induction of adipocyte genes downstream of PPAR γ during adipogenesis. We have also updated the Discussion.

3. The co-IP of BRD4 and CEBP β is not compelling in Figure 4D given a lack of controls and the faint signals for BRD4 and CEBP β in the IP from CRE-treated cells. To address specificity, the authors should probe for another nuclear protein in the BRD4 IP and perform a control IP of another nuclear protein. These experiments will determine whether the association between BRD4 and CEBP β is specific or a result of general enrichment for transcription factors and/or chromatin in the IP fraction.

We have repeated the co-IP of Brd4 and C/EBP β in WT and Brd4 KO cells during adipogenesis (new Figure 4d). RbBP5 IP and CDK9 IP were included as negative and positive controls, respectively.

4. In Figure 3, the authors report changes to the BRD4 cistrome (panel A) and abundance (panel B) during adipogenesis. Can the authors provide insight into whether the changes in BRD4 occupancy result from genomic redistribution of BRD4, decreased abundance of BRD4 or both? Heatmaps showing read counts at all binding sites will be more informative than Venn diagrams that rely on hard cut-offs.

Since this point was raised by both Reviewers #2 (see point 1) and #3, we have reexamined Brd4 protein levels during adipogenesis using nuclear extracts. We repeated the Western blot twice using nuclear extracts from 2 different cell lines and found that Brd4 levels decrease mildly at D2 of adipogenesis. These new Western data are consistent with the qRT-PCR data of Brd4 shown in Figure 2e. We have replaced the original Western blot data obtained from whole cell lysates with the new data obtained from nuclear extracts (the new Figure 3b). As suggested by Reviewer #3, we have generated heatmaps of all Brd4 binding sites during adipogenesis and included it in the new Figure 3c. This new data clearly shows the redistribution of Brd4 during adipogenesis and provides more useful information.

Minor concerns:

5. Figures 4A-C give essentially the same results as figures 5C-E. Some of these data can be moved to the supplemental information to streamline the paper without a loss of information.

We moved Figures 5c-e to a new supplementary Figure 6 and updated results and figure legends accordingly.

6. More details on the motif mining approach used for Figure 4A should be included in the methods. It is unclear as to which bioinformatic tools were utilized, which genomic regions were used for the background and whether the background properly compensates for differences in nucleotide composition/motif prevalence in promoters vs enhancers. This is particularly important given that the proportion of binding sites at enhancers versus promoters changes dramatically over the adipogenic time course used to map Brd4 occupancy.

We have described the motif search tool in the methods section. We used “SeqPos” tool in Galaxy Cistrome⁹. This tool uses the distances from motif positions to the peak summits (center of peaks) to find the most enriched motifs and does not require user-provided background sequences as input. Detailed algorithm of this tool has been described previously¹⁰.

7. Scale bars should be included with the micrograph images of Figure 1D.

We have added a scale bar in the Figure 1d.

8. Page 5, line 90: “Consistently, knockdown of Brd4 in 3T3-L1....” The use of the word “consistently” is confusing. Does it mean that the knockdown experiment worked every time, or that the knockdown produces a phenotype consistent with the genetic knockout? This should be rephrased to communicate the correct idea unambiguously.

We have modified the sentence to clarify.

References

- 1 Yao, X. *et al.* The Mediator subunit MED23 couples H2B mono-ubiquitination to transcriptional control and cell fate determination. *EMBO J* **34**, 2885-2902, doi:10.15252/embj.201591279 (2015).

- 2 Deato, M. D. *et al.* MyoD targets TAF3/TRF3 to activate myogenin transcription. *Mol Cell* **32**, 96-105, doi:10.1016/j.molcel.2008.09.009 (2008).
- 3 Roberts, T. C. *et al.* BRD3 and BRD4 BET Bromodomain Proteins Differentially Regulate Skeletal Myogenesis. *Scientific reports* **7**, 6153, doi:10.1038/s41598-017-06483-7 (2017).
- 4 Efroni, S. *et al.* Global transcription in pluripotent embryonic stem cells. *Cell Stem Cell* **2**, 437-447, doi:10.1016/j.stem.2008.03.021 (2008).
- 5 Roe, J. S., Mercan, F., Rivera, K., Pappin, D. J. & Vakoc, C. R. BET Bromodomain Inhibition Suppresses the Function of Hematopoietic Transcription Factors in Acute Myeloid Leukemia. *Mol Cell* **58**, 1028-1039, doi:10.1016/j.molcel.2015.04.011 (2015).
- 6 Bhagwat, A. S. *et al.* BET Bromodomain Inhibition Releases the Mediator Complex from Select cis-Regulatory Elements. *Cell Rep* **15**, 519-530, doi:10.1016/j.celrep.2016.03.054 (2016).
- 7 Jang, M. K. *et al.* The bromodomain protein Brd4 is a positive regulatory component of P-TEFb and stimulates RNA polymerase II-dependent transcription. *Mol Cell* **19**, 523-534, doi:10.1016/j.molcel.2005.06.027 (2005).
- 8 Ge, K. *et al.* Alternative Mechanisms by Which Mediator Subunit MED1/TRAP220 Regulates Peroxisome Proliferator-Activated Receptor γ -Stimulated Adipogenesis and Target Gene Expression. *Mol. Cell. Biol.* **28**, 1081-1091 (2008).
- 9 Liu, T. *et al.* Cistrome: an integrative platform for transcriptional regulation studies. *Genome Biol* **12**, R83, doi:10.1186/gb-2011-12-8-r83 (2011).
- 10 He, H. H. *et al.* Nucleosome dynamics define transcriptional enhancers. *Nat Genet* **42**, 343-347, doi:10.1038/ng.545 (2010).

REVIEWERS' COMMENTS:

Reviewer #2 (Remarks to the Author):

This reviewer is satisfied with authors' responses. Congratulation on a nice piece of work!

Reviewer #3 (Remarks to the Author):

Lee and colleagues nicely revised their manuscript to meet the concerns of this reviewer and those of the other reviewers. Well done.